# Possible Bias in Supervised Deep Learning Algorithms for CT Lung Nodule Detection and Classification

**DOI:** 10.3390/cancers14163867

**Published:** 2022-08-10

**Authors:** Nikos Sourlos, Jingxuan Wang, Yeshaswini Nagaraj, Peter van Ooijen, Rozemarijn Vliegenthart

**Affiliations:** 1Department of Radiology, University Medical Center of Groningen, 9713 GZ Groningen, The Netherlands; 2Department of Radiation Oncology, University Medical Center Groningen, 9713 GZ Groningen, The Netherlands

**Keywords:** pulmonary nodules, AI, deep learning, lung cancer, detection, classification, bias, validation, chest CT

## Abstract

**Simple Summary:**

Artificial Intelligence (AI) algorithms can assist clinicians in their daily tasks by automatically detecting and/or classifying nodules in chest CT scans. Bias of such algorithms is one of the reasons why implementation of them in clinical practice is still not widely adopted. There is no published review on the bias that these algorithms may contain. This review aims to present different types of bias in such algorithms and present possible ways to mitigate them. Only then it would be possible to ensure that these algorithms work as intended under many different clinical settings.

**Abstract:**

Artificial Intelligence (AI) algorithms for automatic lung nodule detection and classification can assist radiologists in their daily routine of chest CT evaluation. Even though many AI algorithms for these tasks have already been developed, their implementation in the clinical workflow is still largely lacking. Apart from the significant number of false-positive findings, one of the reasons for that is the bias that these algorithms may contain. In this review, different types of biases that may exist in chest CT AI nodule detection and classification algorithms are listed and discussed. Examples from the literature in which each type of bias occurs are presented, along with ways to mitigate these biases. Different types of biases can occur in chest CT AI algorithms for lung nodule detection and classification. Mitigation of them can be very difficult, if not impossible to achieve completely.

## 1. Introduction

Lung cancer is one of the deadliest forms of cancer [1], the reason being that most of these cancers are detected in late stages. Lung nodules could be a sign of lung cancer and their early detection combined with early treatment significantly reduces lung cancer mortality [2,3]. Radiologists are responsible for detecting these nodules as part of their daily routine. These nodules can be most clearly identified in thin-slice chest CT scans. The reason why chest CT is preferred over any other modality is that it has higher sensitivity for the detection of lung abnormalities, including small nodules [4]. Therefore, many AI algorithms use chest CT data to detect, segment, or classify lung nodules. Given that many scans need to be examined, nodule identification by the radiologists in chest CT is error-prone, and potentially harmful nodules may be missed due to the heavy workload [5]. Many pilot studies have demonstrated the potential of lung cancer screening in the detection of early-stage lung cancers and future implementations of such screening programs would result in a huge increase in the number of CT scans that need to be reviewed [6].

In recent years, AI algorithms that can assist radiologists in their daily workflow have gained traction. There are many potential uses of such software [7]. In lung cancer screening, one of these uses is to identify normal scans for which the algorithm is confident that no nodules exist. Then, the algorithm will exclude these normal scans from the list of scans that need to be reviewed, leaving only those which contain nodules to be further examined by the human reader [7]. Using an algorithm in such a setting may help in significantly reducing the burden and fatigue of radiologists [8]. In addition to the above, AI algorithms can be used in triage mode to sort scans based on the suspiciousness or acuteness of the potential findings, to ensure that radiologists will first examine the most relevant scans. Moreover, such software can be used in medical image interpretation as a first or second reader [9]. The potential of using an AI tool for concurrent/second reading of chest CT scans was demonstrated in a study by Muller et al. [10]. This study resulted in finding additional nodules of interest without increasing the reading time of the scans by the radiologists. Apart from these uses, AI algorithms can also be utilized to detect the exact location of nodules, segment them and quantify their size. There are many AI software packages that have already been approved by FDA that perform such tasks [11]. In addition to that, another potential application of AI algorithms is to determine a risk score to classify nodules as (more likely) benign or malignant. However, the development of such algorithms is still under development [12]. In all the above tasks, it is essential for the AI algorithm to decide with high confidence if a pulmonary nodule is present in a scan or not. In Figure 1 a flowchart of the lung cancer screening workflow is presented, along with the steps in which AI can be used to aid clinicians.

With any AI algorithm developed, there is a possibility that it contains bias that will influence the decisions of the radiologists [13], and eventually, the benefits versus harms of lung cancer screening. Bias can be thought of as the errors in the results of an AI algorithm for a given task that creates unfair outcomes. Because of the high dependence of AI on the training data, caution should be given to the selection and composition of the database. Shortely, for medical imaging applications, bias could be reflective of the specific population subgroup that the algorithm was trained on, and the image parameters used to acquire the data [14,15]. For example, if an AI algorithm was trained on a screening cohort, which in general contains mostly healthy individuals, it may not be adequate to be used in daily clinical practice in which there may be many incidental nodules in higher risk and diseased patients. Moreover, datasets used to develop these algorithms may reflect socioeconomic status inequalities and may discriminate based on sex, race, age, etc., leading to undesired outputs for specific groups.

Caution should also be given to the selection of metrics to evaluate the performance of AI algorithms since most of them may not be appropriate and may provide wrong information that, in turn, may result in a biased estimate of their performance [7,16]. There are published papers that provide a detailed overview of the different sources and types of bias in AI in general, like by Mehrabi et al. [17] and Suresh et al. [18], but there is no published work focusing on bias in AI algorithms applied in Chest CT.

In this paper, we identify and explain the potential bias in medical imaging AI algorithms, with special focus on lung nodule detection and classification in chest CT. We also present ways proposed in the literature to mitigate the effects of most of the occurring biases.

## 2. Materials and Methods

### 2.1. Overview

This literature presented in this work was collected from many different journals, most found through PubMed. Since this material was mainly extracted from ‘Discussion’ and/or ‘Limitations’ sections of articles found in the literature, this review was not conducted systematically. It rather contains most of the biases mentioned in chest CT imaging AI papers published from 2010 until the present day, and which are related to chest CT nodule detection and classification.

### 2.2. Types of Bias in Chest CT AI Algorithms

There are many different phases in model development for lung nodule detection and evaluation in chest CT, as shown in Figure 2.

For these different phases, there are many different types of bias in AI algorithms known from the literature [17]. Figure 3 lists the most common ones in chest CT for lung nodule detection and classification.

1*Collider bias:* A Collider is a variable that is the effect of two other variables. If it is assumed that these two variables influence how samples are selected, the association between these variables may become distorted, affecting the collider directly or indirectly. If then, the collider is used to draw conclusions about these associations, these conclusions will be invalid [19].2*Cognitive bias:* In the training dataset used to develop an algorithm, radiologists are commonly responsible for finding and annotating nodules. If most radiologists missed a particular nodule and if a majority vote is used to decide if this finding is a nodule or not, then the resulting annotation will be incorrect for that finding (erroneously classified as non-nodule). The difference in interpretation and diagnosis by different radiologists results in cognitive bias [13,20].3*Omitted variable bias:* When creating an AI algorithm, some important variables that can affect its performance may be deliberately left out, leading to this type of bias [21].4*Representation/Racial bias:* Representation bias occurs when the dataset used to develop an AI algorithm is not diverse enough to represent many different population groups and/or characteristics. Sometimes, this type of bias is indistinguishable from racial bias; this could significant harm the underrepresented group [16].5*Algorithmic Bias:* Most biases are assumed to be unintended, meaning they are unwanted and appear mostly during the data collection and data selection process. Algorithmic bias is an example of bias inserted on purpose and can be the result of the motives of the programmers/companies that develop the algorithm [22]. An AI algorithm can be deliberately designed to skew results f.e. with the goal to maximize profit.6*Evaluation bias:* Such bias arises when an evaluation of the performance of an algorithm is conducted using an inappropriate dataset, like an internal one, which resembles the data used to train the algorithm, or even a contaminated evaluation dataset that contains samples from the training dataset [17]. This results in an overly optimistic estimate of the performance of the algorithm, compared to when this evaluation is performed using an external dataset collected, for example, from another hospital.7*Population bias:* If an AI algorithm is developed using data acquired under specific conditions e.g., a specific population, and is applied to different clinical settings than those it was developed for, this type of bias emerges [17].8*Sampling bias:* This bias can occur when an AI algorithm is developed using data sampled non-randomly, which may miss important cases and features of interest encountered during clinical practice [15].9*Publication bias:* A common type of bias in many fields. This bias results from journals not willing to publish studies with negative or suboptimal findings, or studies that confirm or reject already published results (replication studies) [23].

## 3. Results

### 3.1. Bias in Chest CT Nodule Detection and Classification


*Collider Bias*


To give an example of this bias, consider the case of a general hospital in a region, where a chest CT screening trial is implemented. Let us assume that most people who live in the nearby region do not fulfill the eligibility criteria to participate, usually because they are low-risk individuals. This could be the case if students are the majority of people living near the hospital. Consequently, most screenees will go to that hospital from far away from where the hospital is located. Therefore, taking part in the chest CT screening program is our collider. This means that a person may take part in the study because it lives in the surrounding area of the hospital and fulfills the eligibility criteria (is above a certain age and in high-risk), or because it belongs to a high risk group, from a region far away from the hospital. Even though the two causes are independent, it seems to be a relationship between belonging to a high risk group and living in a region far away from the hospital. Those who live far from the hospital usually participate in the study and belong to high-risk groups whereas those who live close by do not. This correlation is the result of conditioning on a collider and having in the dataset only patients from one hospital [24]. Therefore, if the patient’s address is used as a feature to train an AI algorithm that predicts if a person is at high risk of having lung nodules, this algorithm will be biased and most of the times will wrongly classify individuals who live near the hospital as being in low risk.


*Cognitive Bias*


An example of a potential bias in chest CT could exist in the study of Cui et al. [25]. In this study, many low dose CT scans from participants with different nodule and population characteristics were used to train an algorithm for lung nodule identification. Even though the reference standard/ground truth used in this study was provided by the agreement of three experienced radiologists (more than 20 years of experience), it might be possible that they missed some nodules (human error). Then, nine radiologists (five seniors with more than ten years of experience and four juniors with more than five years of experience) independently identified nodules in these scans, so that their performance will be compared to that of the AI algorithm, their sensitivity ranged from 64–96%, with an average of 82% [26]. In the same study, it may also be the case that the three experienced radiologists who provided the ground truth incorrectly classified a nodule contained in the LIDC/IDRI dataset (wrong diagnosis). This may lead to wrong conclusions about the performance of an AI algorithm and introduce bias to it due to the incorrect labels that exist in the data since this algorithm will be assessed based on the fact that a nodule is a normal region and vice versa.

In addition to the above, a study by Deveraj [27] showed that areas in the chest like the endobronchial, hilar, and paramediastinal region are those in which radiologists tend to miss most of the nodules on screening CT scans. Similarly, in a study by Veronesi et al. [28], lesions found endobronchially and in the center of the lung were those that tend to be missed more often. Therefore, nodule location is also a factor that affects the performance of radiologists in detecting nodules [29].


*Omitted Variable Bias*


One example of this type of bias can be found in [30], where the effect of radiation dose, patient age, and CT manufacturer on the performance of an AI nodule detection algorithm was explored. Chi squared testing was used to assess if the performance of the AI algorithm is dependent on these factors. In particular, expected values in a two-way table for each of those variables were estimated and used to calculate the test statistic. The detection sensitivity was not affected by dose (x^2^ = 1.1036, *p* = 0.9538), and was independent of patient age (x^2^ = 6.1676, *p* = 0.8010) and of CT manufacturer (x^2^ = 10.5136, *p* = 0.7862). Moreover, the model had higher detection sensitivity for solid nodules >6 mm and calcified nodules compared to smaller, and non-calcified nodules. This is the result of more abundant features contained in larger nodules and of higher signal intensity of calcified nodules in CT images. Although the effect of age on the performance of the algorithm was examined, smoking history was not investigated, which may be a confounding factor since nodule characteristics like type and size differ by smoking status [31]; furthermore, smoking is one of the most well-known and prevalent risk factors for lung cancer [32].

Another example when this type of bias appears is when patients with COVID are excluded from a collection of datasets for lung cancer screening. CT findings of COVID infection may resemble other disorders [33], and these findings may be falsely classified by the algorithm as nodules leading to more false positives and therefore, to a decreased performance. An example of such a case is shown in Figure 4. Similarly, omitting the presence of other diseases like emphysema could also influence the number and characteristics of nodules [34] and so, the performance of that algorithm.


*Representation/Racial Bias*


An example of this type of bias can be found in the recent work of Banerjee et al. [35] which showed that self-reported race can be easily predicted by AI algorithms in datasets of multiple imaging modalities (including chest CTs), just by using the image pixels as input in these algorithms. The performance of these AI algorithms was robust in the external validation that was also performed. These results cannot be attributed to confounding variables such as age, tissue densities, etc. of each race. In addition to that, the learned features involved all regions of the images that were used as an input to the system. Therefore, attempts to mitigate these biases will not be trivial. This work also showed that AI models have the intrinsic capability to learn to discriminate based on information about the race that is trivially learned by them, even from noisy and corrupted images. These images may even be extremely degraded to the point that they cannot be recognized as medical images by human experts. Therefore, the race of individuals based on the same images that were used as input to the algorithm cannot be recognized, and so, it will not be possible for humans to recognize if the AI system is discriminatory just by looking at those images. It may be possible that the algorithm will base its output on race, and e.g., may impact healthcare access to patients of a particular ethnic background.

Another example of this type of bias is related to disease prevalence which has a great variation in different populations [25]. If the dataset used to develop the algorithm has a higher disease prevalence than the population in which it will be used in practice, the model may become oversensitive and detect nodules that may not be of clinical significance [25]. These additional findings will most likely be false positives leading to a reduction in the performance of the algorithm.


*Algorithmic Bias*


A hypothetical example of such a case could be a system that leads more patients to undergo simple follow-up scans for potential findings without needing to, which in turn leads to more profit for hospitals/companies that provide the equipment and the software. This bias can also be introduced in the way the AI algorithm uses patient data. For example, it may recommend further checks only in those patients that can afford to pay or based on their insurance status [22].


*Evaluation Bias*


One other very common type of bias in algorithms for lung nodule detection based on chest CT data is evaluation bias. This bias occurs because there are only a few publicly available manually annotated datasets with the coordinates of lung nodules that can be used to train an AI nodule detection algorithm. In general, most studies that report metrics on the performance of AI nodule detection algorithms in chest CT scans were trained and validated using either the LUNA16 or the LIDC-IDRI dataset (of which LUNA16 is a subset) [25,26,36]. Therefore, most of the published results have not been validated using an external dataset and the developed systems are likely to be biased towards the specific characteristics of the scans of the LUNA16/LIDC-IDRI dataset.

There are also examples of algorithms developed and tested using a few external datasets. In the study of Hosny et al. [37] authors used 7 independent datasets across 5 institutions of 1194 patients in total of non-small-cell lung cancer CT images to develop a tool for mortality risk stratification. Even though the dataset is heterogeneous, the sensitivity of the used AI algorithm under different clinical and image acquisition parameters, like tube current, reconstruction kernels, etc., was not assessed. Thus, it cannot be ensured that the CAD system is not biased towards the specific characteristics of these parameters.

Usually, when authors refer to ‘validation’ in their studies, they mean that they kept a subset of their dataset out to be used as a test set to assess the performance of their AI algorithm. The presented metrics may then represent an overestimation of the true performance of the algorithm, compared to if it was validated using an external dataset. An example of such a case is presented in Gruetzemacher et al. [38].


*Population Bias*


Sometimes an algorithm may only be developed and tested with a dataset of individuals that undergo screening. That means that all individuals who fulfill the screening criteria and from whom a CT scan was obtained, are included in that dataset. This population may not be representative of those encountered during clinical practice with incidental lung nodules since the characteristics of a screening population are different from patients encountered in daily clinical routine. For example, an algorithm developed using scans from the NLST screening trial [39] may not be used to identify nodules in daily clinical practice before it is ensured that it can be generalized to that setting. The bias that is related to having a training set with only individuals from a specific population that is different from the one in which the algorithm will be applied to is called selection/population bias [40].


*Sampling Bias*


Sampling bias can occur, among others, if nodule selection to develop an AI algorithm was performed in a non-random way and could result in not taking into consideration some important nodule characteristics in the creation of a training set. A potential example of such bias may be present in Wang et al. [41] in which an algorithm that automatically classifies subsolid nodules in CT images as atypical adenomatous hyperplasia (AAH), adenocarcinoma in situ (AIS), minimally invasive adenocarcinoma (MIA), or invasive adenocarcinoma (IAC), was developed. This algorithm could help differentiate the degree of malignancy of these nodules. Even though the results were promising, further validation should be performed since the size of those nodules was not considered, although it is one of the most important/influential factors that indicate malignancy [42]. One of the preprocessing steps required before feeding a volume to that algorithm is resizing, which may result in loss of information of the actual size of the nodule. This could happen because lesion features that play a major role in the classification may be discarded and lead to incorrect classification outcomes. If the algorithm is trained using patches of volumes in which information about nodules is lost due to resizing, it will likely be biased and performs well only on nodules in which resizing does not result in information loss. A way to check for the presence of such bias is to perform external validation on many different nodule sizes before implementing such an algorithm in hospitals.

Lastly, another example of sampling bias can be found in the way the LIDC/IDRI dataset was acquired. Most of the scans in that dataset were acquired using a GE scanner (896), whereas there are only a few scans acquired with Siemens (234), Philips (74) and Toshiba (69) scanners [43]. Consequently, the developed algorithm may have degradation in its nodule detection performance when this is assessed in an external test set that consists of scans of the minority vendors of the training dataset. For the specific case of vendors, this is called vendor (or single source) bias [44].


*Publication Bias*


In the published literature, there are many studies that assess the performance of CAD systems. In a systematic review of studies in which the performance of these algorithms was tested on a dataset not derived from LIDC-IDRI [45], the authors noticed that there is a high risk for the aforementioned bias to be present, since those who conducted these studies and got negative or suboptimal results compared to the already published ones, may not submit their work to be published. In general, algorithms developed for a specific task which are less biased but do not perform as well as already published ones, will likely fail to be published with a high impact factor. In addition to the above, in a systematic review conducted by Huang et al. [46] on studies that assess the diagnostic performance of AI algorithms in the classification of pulmonary nodules it was shown that there is a risk of publication bias in these studies, based on Deek’s asymmetry test (*p* < 0.05).

### 3.2. Ways to Mitigate Bias

Different types of bias could appear in any stage of AI model development. To recognize if the developed model contains any form of bias, each stage has to be carefully examined. Bias mitigation is, in most cases, an extremely challenging task. Even though some methods have been proposed to deal with some of these biases, there is not a common agreement on which method is preferable and there is no guarantee that any of these methods will work under specific circumstances.

The procedure of understanding what biases might be contained in an AI algorithm starts by inspecting misclassified examples to check if there are any patterns that could be identified in them. The most desired solution to deal with bias and find these patterns is model explainability. This can help us to better understand how an AI model makes its decisions so that the AI algorithm can be trusted by physicians. This method provides a detailed overview of how the decision of the AI model was made and of how the model arrived at a specific result. This explanation should be described in natural language so that it can be understood by humans. Even in cases in which the algorithm provides wrong results, by looking at its decision-making process, one could be able to identify what caused the error. In practice, one can only seek interpretability. For AI models interpretability means knowing the regions of an image that played a major role in the classification result. An example of a method that can be used towards achieving interpretability is to create lesion localization (heat) maps. These heat maps can help in the interpretation of the algorithm’s classification results, e.g., as shown in Lee et al. [47]. An example of a heat map (Grad-CAM method) is shown in Figure 5 [48,49]. By using these maps, it can be assessed which scans the algorithm classifies correctly as positives for the wrong reasons, and which scans are missed by the algorithm, which can also be used as indications for possible biases [50]. Examples of biases that could be possibly identified with this method are the representation bias, the evaluation bias, the population bias, and the sampling bias. Unfortunately, often times heat maps may not be adequate for the goals of bias detection since this method does not integrate the reasoning of the decision, meaning that they do not provide information of the factors that lead to the classification results.

Since most deep learning algorithms are like a black box, interpretability is extremely difficult (if not impossible) to be achieved and explainability by attention maps is not available for all the current AI algorithms. Therefore, a better approach is to seek reviewability in which we do not necessarily have explanations, but we expose the decision-making process including human processes, structures, and systems around a model [51]. Reviewability involves exposing information about context, decisions of an algorithm for legal compliance, whether it operates within expected/desired parameters, etc. In some cases, it might be appropriate to give explanations and act in line with some principles throughout the decision-making process, as well as provide information on the evaluation procedures from those that deployed these algorithms, the decision of engineers who developed the system, data used in training and testing, information about the effects, fairness, and lawfulness of those algorithms in practice. Each domain may have different requirements. Examples of biases that could be possibly identified with this method are the collider bias, the cognitive bias, the omitted variable bias, the representation bias, the evaluation bias, the population bias, and the sampling bias.

In practice, what has been proposed is a standardized framework to present the whole training process of an algorithm, its parameters, information about the training/possible validation dataset used, etc. This could inform the end-users about the limitations and even the biases that may exist in that algorithm and advise them under which settings the algorithm should be used in practice. Gebru et al proposed the use of ‘datasheets for datasets’ to elucidate and standardize information about public datasets, or datasets used in the development of commercial AI services and pretrained models [52]. Information that should be included in this datasheet should be provenance, key characteristics, relevant regulations as well as potential bias, strengths, weaknesses, and suggested uses of the aforementioned datasets [14]. Another way to present details about the whole creation pipeline of an AI model is called FactSheets. This contains information about the performance of an algorithm, safety, security, and provenance and is suggested to be completed by the creators of the software to inform the consumers [14]. In Box 1 a list of possible questions is designed using the above methods which can be used to identify possible sources of bias. In general, AI model evaluations in published literature are often poorly performed [53] and do not provide the information required for clinical assessment. Therefore, adopting these frameworks could be extremely beneficial. It is also worth mentioning that one other way to mitigate unwanted biases in a dataset is through adversarial learning, such as the one presented in [54].

Box 1Questions that an AI model developer could use to identify possible sources of bias.
-How was the collection and annotation procedure of the dataset performed? Who contributed to that and on behalf of which entity?-Which characteristics/parameters were considered in the dataset creation process? How many examples of each of those characteristics/parameters were included in the dataset?-For which population groups and/or characteristics should the developed algorithm be used? Is the dataset representative of that population?-What are the characteristics of the dataset used for model validation?-Who funded the dataset creation process?-Was any preprocessing or cleaning of the dataset performed?


In addition to the above, it should also be mentioned that quite often all the above methods may fail to provide solid evidence about the existence of bias when using an AI tool. The only way to limit as much as possible the presence of bias in these tools is to perform software validation before using a specific software package in clinical practice. This validation should be performed using an external dataset, which ideally is multicentered, and contains a lot of cases for which the existence of bias should be assessed. For example, if the goal is to check if a software package is biased towards a specific gender or race, the performance of this software should be evaluated with individuals of both genders and for a wide range of races. Such tasks are of critical importance, especially in the case of commercial software packages in which, most of the time, the characteristics of the training set used to develop the AI tool are unknown. It is also essential to monitor these tools when used in clinical practice on a daily basis to ensure that there are no failure cases and/or biased results.

At last, it should also be pointed out that the sources of bias are not limited to those that were mentioned in the previous sections. There are also many other sources of bias which may or may not be possible to mitigate. For some of them, it may not be possible to even recognize their presence. In general, a few guidelines to deal with undesired sources of bias are to have bigger sample sizes (to recognize possible outliers), have better designed protocols and methods for image acquisition (improved image quality), and observe the performance of the algorithm once it is deployed in clinical practice to ensure that bias is not introduced from cases that the algorithm was not trained on [55]. Sometimes, it may also be desired to have some bias in an AI algorithm since in order to achieve generalization in new examples implicit assumptions should be made f.e. about the underlying data distribution.

## 4. Discussion

In this literature review, we summarized the main potential sources of bias in AI algorithms for lung nodule detection and classification in chest CT. To determine if an AI algorithm can accurately find all nodules in a CT scan with a few false positives, manual annotations in different circumstances should be evaluated (different imaging parameters, nodule shapes, sizes, etc.) [56]. Radiologists’ sensitivity depends on nodule size [57] and therefore, depending on how the AI algorithm is intended to be used (first, or second reader, screening, etc.), it may perform better compared to radiologists. In general, even with an agreement between radiologists to solve discrepancies, there are limits to how well images can be annotated which also sets a ceiling on the performance of the AI algorithm. This limit is also directly related to the presence of a gold standard like biopsy results which are critical to further verify the performance of an AI algorithm, as well as to confirm the findings of the radiologists. Unfortunately, most of the times this gold standard is absent and is difficult if not impossible to be obtained.

Moreover, it is worth mentioning that any AI algorithm created to perform lung nodule detection and/or classification will likely have at first both false positives and false negatives. There is always a trade-off between these two. If the algorithm is designed such that it does not have any false negatives, the number of false positives will increase and vice versa. When the AI model misses nodules, and if the scan will not be further reviewed by a radiologist, there is a possibility that an individual will be erroneously considered as healthy. This may lead to a delayed diagnosis or even to an unfavorable prognosis. On the other hand, if the algorithm has many false positives it may consider the individual as being at high risk and a radiologist will have to prioritize the check of that scan, leading to a non-proper allocation of resources since the radiologist may not review more severe cases due to limited time and/or it may increase reader fatigue. Depending on the specific task that the algorithm will be applied to, and based on the feedback of radiologists, proper thresholds can be set for the predictions to achieve the best balance in performance, based on what is considered as most important to optimize, precision or recall. Even if this is performed, there is still a possibility to have confounding variables that may affect the results. It is therefore essential to eliminate the effect of as many confounding variables as possible and to prevent the introduction of biases by them.

In general, bias in AI algorithms occurs very often. Algorithms developed by companies for commercial use should get FDA approval in the USA and/or CE approval in Europe before they are allowed to be used in clinical practice. A detailed list of all commercial algorithms that have received a license and that are focused on chest CT scans can be found in [11]. Even after that approval, there is no guarantee that these algorithms will not contain bias. Therefore, the FDA requires that the performance of these tools will be evaluated when used in clinical practice and feedback should be provided to the company that developed these tools to improve its performance and mitigate possible bias. Similar guidelines for the ethical use of AI have been proposed by the EU [58]. One of the points in these guidelines is the need for human oversight of AI algorithm results to ensure that these algorithms work according to their designed specifications. Monitoring these systems is essential in improving their results and avoiding future bias.

In addition to the above, when deep learning algorithms are implemented in clinical practice, a special infrastructure will likely be required. That infrastructure could be specialized GPUs in a server in the hospital or access to a cloud platform for the required computations. Moreover, the whole data exchange should be performed in a secure way to protect patients’ privacy. This specialized infrastructure could only be implemented in hospitals that have the available resources to buy and support it. In case that an AI algorithm’s usage is proven to improve clinical outcomes the infrastructure required to run should be as easily accessible as possible.

It is also worth noticing that even if most of the bias could be mitigated there will still be examples in which the wrong decisions are made by an AI algorithm. In these cases, there are many ethical and legal dilemmas, such as who is responsible for these mistakes? Is it the doctor who made the final diagnosis, the programmer who created the AI algorithm, the company that sells it, or someone else? These questions remain unanswered even today. There is a need for further discussion about the laws and regulations that should be established to prevent misuse of these tools and to deal with the possible consequences of their wrong outputs.

At last, since most AI algorithms only use image data as input, there is the need to also incorporate clinical information into these algorithms, like smoking status, in particular where it concerns nodule risk calculators of malignancy.

This study also comes with some limitations. One of the limitations is that it was not possible to conduct this review systematically since bias could only be identified mostly as part of the ‘limitations’ and ‘discussion’ sections of each paper. Our review is, therefore, not exhaustive. Another limitation is that even though we addressed many possible biases, it is not possible to suggest ways to counter all of them. A few good ways to begin with are to have bigger sample sizes in the dataset, have better protocols for the acquisition of the images of the datasets and regularly check the performance of an algorithm when it is deployed and used in clinical practice [55]. Furthermore, it should also be noted that the task of distinguishing between different types of bias is challenging. There may be overlap between the different categories and so, examples presented in this work may fall into different or multiple categories, depending on how each category is interpreted.

In addition to the above, for most of the commercial algorithms, the code used to train an AI algorithm (parameters, architecture, etc.) and the dataset used are not publicly available. This significantly limits the ability to compare algorithms and check for bias in them. Until today, there is no benchmark dataset available that can be used to address the sources of bias that exist in each nodule detection and classification system. There is a great need to create such a benchmark dataset. More specifically, for the case of lung nodule detection algorithms on low dose chest CT scans, the ImaLife dataset [59] can be used in the future to check for biases in these algorithms, due to its diversity and its great number of scans. Algorithms are only as trustworthy as the data being gathered and used to develop them. A properly deployed AI algorithm should consider as many biases as possible and compensate for them.

## 5. Conclusions

In this paper, we discussed the different biases that can exist in AI algorithms used to detect and classify nodules in chest CT. We also presented a few ways that can be used to mitigate some of the biases that may arise. To our knowledge, this review is the first that attempts to present the biases that may occur in the implementation of AI algorithms related to detecting and classifying lung nodules in chest CT scans. Only after recognizing the exact sources of bias and their causes, it will be possible to deal with most of them. This will eventually help the incorporation of AI algorithms in medical practice.

## Figures and Tables

**Figure 1 cancers-14-03867-f001:**
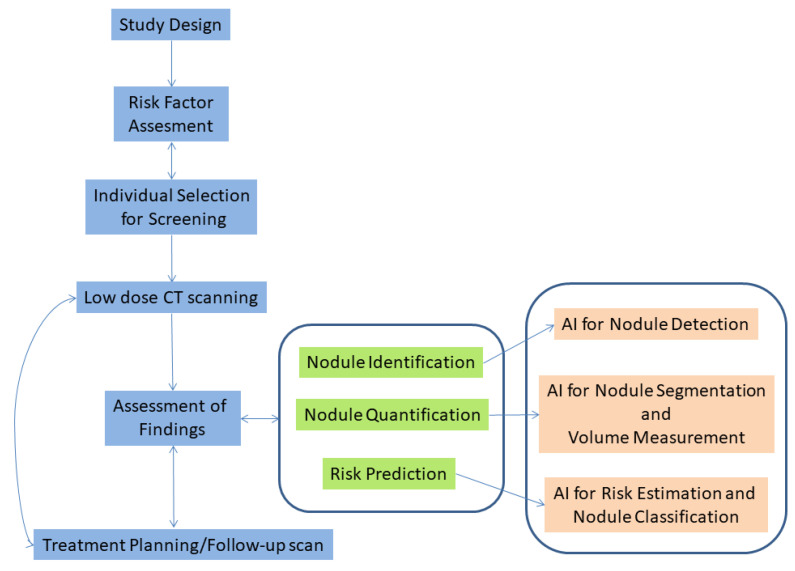
Lung cancer screening workflow along with potential uses of AI software aiming to assist clinicians.

**Figure 2 cancers-14-03867-f002:**
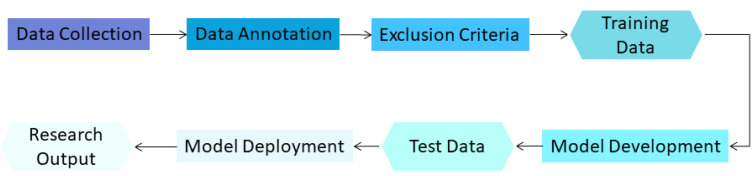
Phases of AI model development in medical imaging.

**Figure 3 cancers-14-03867-f003:**
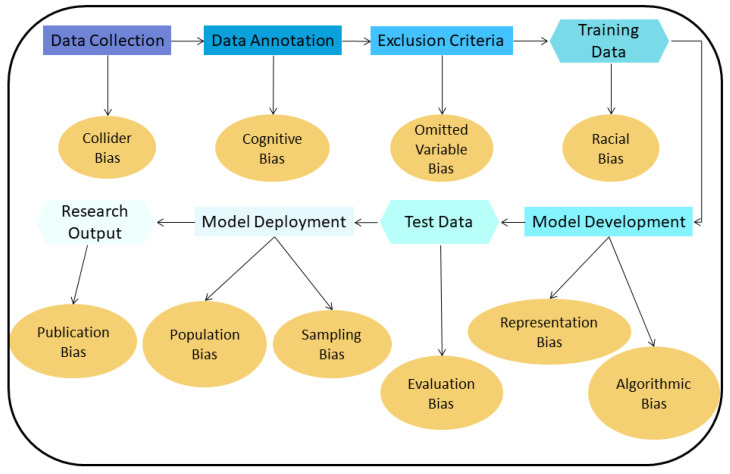
Biases introduced in the different phases of AI model development for lung nodule detection and classification.

**Figure 4 cancers-14-03867-f004:**
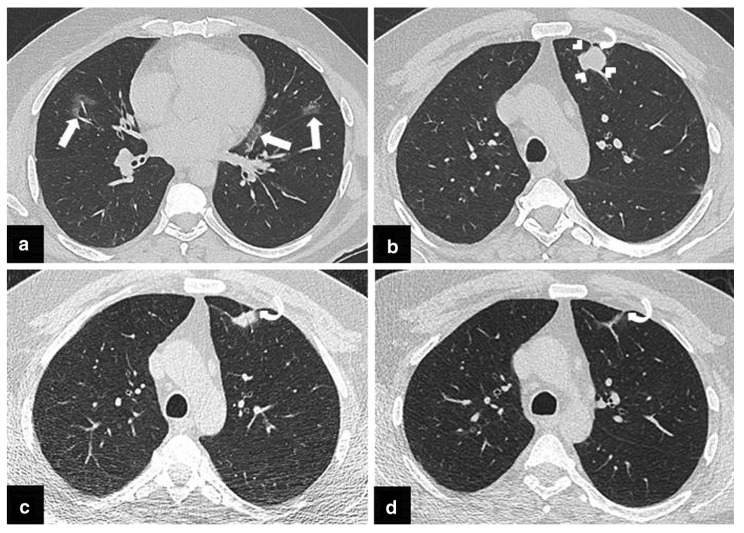
From the publication by Arslan et al. [33] “(**a**,**b**) The initial chest CT scan obtained following PCR test positivity for COVID-19 infection, revealed a few patchy areas of ground glass opacity (GGO) in both lungs (arrows) compatible with COVID-19 pneumonia. An irregularly shaped solid nodule 2 cm in diameter in left upper lobe of the lung was also noted (arrowheads). Percutaneous transthoracic core needle biopsy was scheduled due to suspicion of primary lung cancer. (**c**) CT scan obtained prior to biopsy procedure demonstrated significant size reduction of the nodule. Therefore, biopsy was not performed. (**d**) Follow-up CT scan obtained 3 months later demonstrated complete resolution of the nodule. A pleural tag which became more apparent following resolution of the nodule (curved arrows, (**b–d**) raised the suspicion of COVID-19 triggered focal organizing pneumonia”, licensed under CC BY 4.0.

**Figure 5 cancers-14-03867-f005:**
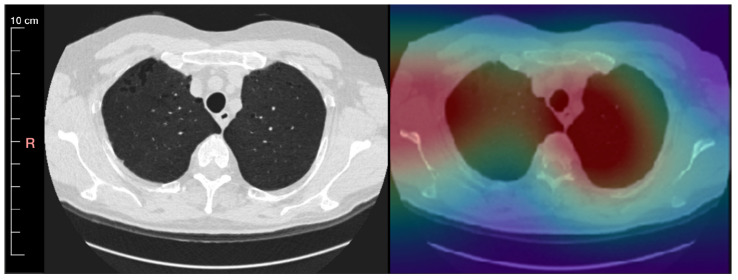
An inspiration chest CT slice of a 65-year-old male patient with mild emphysema is visualized in the lung window level 1500/−500 and reconstructed using medium smooth kernel. Left: CT scan with minimum intensity projection of slab thickness 5 mm Right: A saliency map generated by a convolution of an autoencoder that is overlayed onto the CT image.

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
