# Peer review of "Possible Bias in Supervised Deep Learning Algorithms for CT Lung Nodule Detection and Classification"

_cancers, 2022, doi:10.3390/cancers14163867_

Round 1

Reviewer 1 Report

Dear authors, congratulations for this very interesting paper; in my opinion some improvements are needed. particullary The Collider Bias section should be better explained, in the current version it is not very understandableFurthermore,in the section about on how to avoid or mitigate the various Biases,there is a lack of effective tips that can be used in daily practice. In my opinion the advent of open source algorithms in free AI market is currently not feasible.

Author Response

1) ‘The Collider Bias section should be better explained, in the current version it is not very understandable’:

Thank you very much for your comment. We modified both sections, in ‘Materials and Methods’, and in the ‘Results’. We hope that it is better explained now. You can find the new text below:

‘Material and Methods’: Collider bias: A Collider is a variable that is the effect of two other variables. If it is assumed that these two variables influence how samples are selected, the association between these variables may become distorted, affecting the collider directly or indirectly. If then, the collider is used to draw conclusions about these associations, these conclusions will be invalid

‘Results’: To give an example of this bias, consider the case of a general hospital in a region, where a chest CT screening trial is implemented. Let us assume that most people who live in the nearby region do not fulfill the eligibility criteria to participate, usually because they are low-risk individuals. This could be the case if students are the majority of people living near the hospital. Consequently, most screenees will go to that hospital from far away from where the hospital is located. Therefore, taking part in the chest CT screening program is our collider. This means that a person may take part in the study because it lives in the surrounding area of the hospital and fulfills the eligibility criteria (is above a certain age and in high-risk), or because it belongs to a high risk group, from a region far away from the hospital. Even though the two causes are independent, it seems to be a relationship between belonging to a high risk group and living in a region far away from the hospital. Those who live far from the hospital usually participate in the study and belong to high-risk groups whereas those who live close by do not. This correlation is the result of conditioning on a collider and having in the dataset only patients from one hospital. Therefore, if the patient’s address is used as a feature to train an AI algorithm that predicts if a person is at high risk of having lung nodules, this algorithm will be biased and most of the times will wrongly classify individuals who live near the hospital as being in low risk.

2) ‘In the section about on how to avoid or mitigate the various Biases,there is a lack of effective tips that can be used in daily practice. In my opinion the advent of open source algorithms in free AI market is currently not feasible.’

Thank you for the excellent point. Indeed, the tips provided in that section to mitigate bias are mainly focused on the development phase of the algorithm, as well as before its deployment. We added a paragraph (prelast paragraph above ‘Discussion’ section - “In addition to the above, it should ... and/or biased results.”), hoping to address this concern. Nevertheless, we have to admit that it is still very difficult to check for biases while using an algorithm, and it is better to do that before deploying it in clinical practice. We aim to assess the feasibility (and check for bias) of current AI algorithms for lung nodule detection available in the market in a future publication.

You can find the added text below:

In addition to the above, it should also be mentioned that quite often all the above methods may fail to provide solid evidence about the existence of bias when using an AI tool. The only way to limit as much as possible the presence of bias in these tools is to perform software validation before using a specific software package in clinical practice. This validation should be performed using an external dataset, which ideally is multicentered, and contains a lot of cases for which the existence of bias should be assessed. For example, if the goal is to check if a software package is biased towards a specific gender or race, the performance of this software should be evaluated with individuals of both genders and for a wide range of races. Such tasks are of critical importance, especially in the case of commercial software packages in which, most of the time, the characteristics of the training set used to develop the AI tool are unknown. It is also essential to monitor these tools when used in clinical practice on a daily basis to ensure that there are no failure cases and/or biased results.

Reviewer 2 Report

Dear Authors,

Possible bias in supervised deep learning algorithms for CT lung nodule detection and classification by Sourlos is an interesting review manuscript. The authors apply AI and for automatic lung nodule detection in chest CT scans.

Comments.

1.     As the authors mention that the Artificial Intelligence (AI) algorithms is already in use., then what is the advantage of this new AI system over the ongoing AI?

2.     What is the falls positive detection rate  observed in this new AI system?

3.     Is there any gender or racial disparities can be assessed through this AI

Author Response

1) ‘As the authors mention that the Artificial Intelligence (AI) algorithms are already in use, then what is the advantage of this new AI system over the ongoing AI?’

Thank you for the comment. We assume that there is a misunderstanding here. We did not propose a new AI system. We performed a review on available AI algorithms for lung nodule detection and classification, aiming to identify possible sources of bias in them, and suggest ways to mitigate them. This was done since there are AI algorithms currently used in clinical practice for which there is no evidence that a rigorous validation of their performance was implemented and therefore, they might contain biases.

2) ‘What is the false positive detection rate observed in this new AI system?’

Thank you for your point. As explained above, there is no new AI system proposed here. FP rate is not of importance for the goal of our review and for the algorithms mentioned in our paper.

3) ‘Is there any gender or racial disparities can be assessed through this AI?’

Thank you for your interesting point. In our review, gender and racial disparities could indeed be a point that indicates the presence of bias in an AI algorithm. We refer you to the section ‘Representation/Racial bias’ for more information. We also added a paragraph in the ‘Ways to mitigate Bias’ section (“In addition to the above, it should ... and/or biased results.”) that provides an example of how gender/racial bias could be addressed.

You can find the added text below:

In addition to the above, it should also be mentioned that quite often all the above methods may fail to provide solid evidence about the existence of bias when using an AI tool. The only way to limit as much as possible the presence of bias in these tools is to perform software validation before using a specific software package in clinical practice. This validation should be performed using an external dataset, which ideally is multicentered, and contains a lot of cases for which the existence of bias should be assessed. For example, if the goal is to check if a software package is biased towards a specific gender or race, the performance of this software should be evaluated with individuals of both genders and for a wide range of races. Such tasks are of critical importance, especially in the case of commercial software packages in which, most of the time, the characteristics of the training set used to develop the AI tool are unknown. It is also essential to monitor these tools when used in clinical practice on a daily basis to ensure that there are no failure cases and/or biased results.